# Nitric Oxide Affects Heme Oxygenase-1, Hepcidin, and Transferrin Receptor Expression in the Placenta

**DOI:** 10.3390/ijms24065887

**Published:** 2023-03-20

**Authors:** Patricia Principe, George T. Mukosera, Nikia Gray-Hutto, Ashra Tugung, Ciprian P. Gheorghe, Arlin B. Blood

**Affiliations:** 1Lawrence D. Longo Center for Perinatal Biology, Loma Linda University School of Medicine, 11175 Campus Street, Loma Linda, CA 92354, USA; 2Department of Obstetrics and Gynecology, Division of Maternal-Fetal Medicine, Loma Linda University School of Medicine, 11370 Anderson Street, Loma Linda, CA 92354, USA; 3Department of Pediatrics, Division of Neonatology, Loma Linda University School of Medicine, 11175 Campus Street, Loma Linda, CA 92354, USA

**Keywords:** nitric oxide, placenta, iron nitrosyls

## Abstract

Nitric oxide (NO) is a gasotransmitter that avidly binds both free and heme-bound iron, forming relatively stable iron nitrosyl compounds (FeNOs). We have previously demonstrated that FeNOs are present in the human placenta and are elevated in preeclampsia and intrauterine growth restriction. The ability of NO to sequester iron raises the possibility of the NO-mediated disruption of iron homeostasis in the placenta. In this work, we tested whether exposure of placental syncytiotrophoblasts or villous tissue explants to sub-cytotoxic concentrations of NO would elicit the formation of FeNOs. Furthermore, we measured changes in the mRNA and protein expression levels of key iron regulatory genes in response to NO exposure. Ozone-based chemiluminescence was used to measure concentrations of NO and its metabolites. Our results showed a significant increase in FeNO levels in placental cells and explants treated with NO (*p* < 0.0001). The mRNA and protein levels of HO-1 were significantly increased in both cultured syncytiotrophoblasts and villous tissue explants (*p* < 0.01), and the mRNA levels of hepcidin and transferrin receptor were significantly increased in culture syncytiotrophoblasts and villous tissue explants, respectively, (*p* < 0.01), while no changes were seen in the expression levels of divalent metal transporter-1 or ferroportin. These results suggest a potential role for NO in iron homeostasis in the human placenta and could be relevant for disorders of pregnancy such as fetal growth restriction and preeclampsia.

## 1. Introduction

Pregnancy poses a significant increase in the need for iron due to increased maternal and fetal erythropoiesis as well as placental and fetal growth. As the organ of materno-fetal exchange, placental iron handling is a key determinant of fetal iron availability. At the same time, the placental consumption of molecular oxygen (O_2_) as a function of weight is greater than that of either the mother or the fetus, and thus, it requires large amounts of iron for its own metabolic function. In fact, the placenta appears to guard its own iron stores independently of maternal and fetal iron status, as placental iron concentrations actually increase in the event of maternal iron deprivation [1,2]. Thus, the fetus is particularly vulnerable in the face of dysregulation of placental iron handling.

As is the case for most exchanges between the mother and fetus, placental iron transport is primarily carried out by syncytiotrophoblast cells [3]. Intracellular iron homeostasis is regulated by mechanisms that are well-conserved across various cell types, including syncytiotrophoblasts. The uptake of iron from the maternal circulation into the syncytiotrophoblast cells occurs via the binding of transferrin-bound iron to transferrin receptors (TfR) on the apical plasma membrane or the importing of non-transferrin-bound iron via a divalent metal transporter (DMT1). Within cells, heme oxygenase (HO-1) plays a role in iron homeostasis by recycling heme-bound iron. Intracellular iron, whether it is derived from HO-1, TfR, or DMT1, can be exported across the basolateral membrane into the fetal circulation via ferroportin (FPN) [4], the only known cellular iron exporter. FPN activity is regulated by the hormone hepcidin (HAMP), which can bind FPN and facilitate its degradation in response to increased concentrations of iron in the circulation [5,6,7].

Nitric oxide (NO), a small free radical gaseous molecule that plays a key role in physiological and pathophysiological intracellular signaling [8], binds avidly to both heme-bound and free ferrous iron to form heme-nitrosyls [9], or dinitrosyl iron complexes (DNICs), together known as iron nitrosyls (FeNOs) [10]. This ability of NO to react with labile forms has the potential to disrupt intracellular iron homeostasis [5]. For example, the exposure of cancer cells to NO released from macrophages is associated with the formation of DNICs that are exported from the cell via multidrug resistance-associated protein 1, with the resulting loss of iron contributing to intracellular iron deficiency [5]. In addition to interacting directly with labile iron, NO can also affect iron homeostasis by acting directly on the iron–sulfur cluster of iron regulatory proteins (IRP) [11,12,13,14,15,16]. NO can activate the RNA-binding activity of IRPs, thereby mimicking the effects of iron starvation with resultant effects on the levels of intracellular iron handling proteins such as TfR, DMT1, ferritin, and ferroportin [15]. Thus, FeNO formation in the placenta has the potential to disrupt iron homeostasis, with implications on overall placental function as well as the supply of iron to the fetus.

We have recently demonstrated that the placenta is unique in its capability to form and maintain FeNOs [17,18]. However, the molecular mechanisms underlying the formation of these complexes and their effect on placental function remain unknown. The objective of the current study was to test the hypothesis that the exposure of syncytiotrophoblasts and villous tissue explants to sub-toxic levels of NO, under either normoxic or hypoxic conditions, would result in the formation of FeNO and would also alter the abundance of mRNA and proteins involved in cellular iron homeostasis.

## 2. Results

### 2.1. Determination of NO Donor Dose

Free nitric oxide (NO) has a half-life of only a few minutes in aqueous buffer equilibrated with room air, and thus, to establish a constant and physiological NO concentration over the 24 h treatment period of the study, the slow-releasing NO-donor DETA was employed. Preliminary experiments were conducted to determine the yield of steady-state NO concentrations produced over a 24 h treatment period from DETA at concentrations of 100, 250, 500, 1000, or 2000 μM. Because the rate of NO consumption is dependent upon O_2_ concentrations, the dose-response experiment was carried out under both normoxic (21% O_2_) and hypoxic (2% O_2_) conditions at 37 °C. As shown in Figure 1A,B, NO concentrations were relatively stable over the 24 h period at all doses of DETA. Cell viability assays (Figure 1C,D) determined 500 μM DETA to be the highest concentration that did not result in a marked decrease in cell viability under both normoxic and hypoxic conditions, and thus, subsequent experiments were carried out with 500 μM DETA, which yielded average steady-state NO concentrations over the 24 h treatment period of 208.9 ± 49.1 nM under normoxia and 432.4 ± 209.8 nM under hypoxic conditions.

In aqueous buffer, free NO is oxidized predominantly into nitrite (NO_2_^−^), with a minor fraction being further oxidized to nitrate (NO_3_^−^) [19]. As a result, over the course of the 24 h treatment period with 500 μM DETA, an accumulation of as much as 250 μM nitrite would be expected. While nitrate is thought to be inert in the placenta, we have recently reported that nitrite can be converted into iron nitrosyls (FeNOs) [18], raising the possibility that the nitrite accumulating in the culture media over the course of the 24 h experiment may confound our goal of assessing the effects of NO itself. To control for this possibility, a control group that received 250 μM nitrite was included in addition to the true negative control group, which received 250 μM nitrate.

### 2.2. FeNO Formation in Placental Cell and Villous Explant Culture

To study whether syncytialized trophoblast cells (SCTs) and explants would generate FeNOs if treated with NO, samples were treated with DETA for 24 h. Figure 2 shows the concentrations of NO and its metabolites (NOx; NO, nitrite, FeNOs, and SNOs but not nitrate) in the culture media and cell and explant culture homogenates following the 24 h treatment period. The total NOx (including nitrate) accumulated in the NO-treated culture media (Figure 2A,D) was comparable to that of the nitrite- and nitrate-treated cultures, showing that those groups serve as appropriate controls for the effects of nitrite and nitrate apart from the effects of free NO. In both the lysed SCTs and homogenized explants, we found that the NO-treated group accumulated significantly higher concentrations of total NOx compared to the untreated control. The groups treated with nitrite and nitrate accumulated NOx but not as much as the group treated with NO (Figure 2B,E). Of importance with regard to iron-handling, a further selective assay of only the FeNO levels using PBS/FeCN revealed that FeNO concentrations were significantly increased only in the NO-treated group in both cells and explants (Figure 2C,F), suggesting that free NO, as opposed to nitrite or nitrate, is metabolized to FeNOs in placental tissues.

### 2.3. HO-1, Hepcidin, and TfR Levels Change in Response to NO

The incorporation of iron into FeNOs has the potential to impact cellular iron metabolism. Thus, we next studied the mRNA and protein expression levels of six iron handling genes: heme oxygenase (HO-1), hepcidin (HAMP), transferrin receptor (TfR), divalent metal transporter 1 (DMT-1), and ferroportin (FPN) in response to NO treatment in cell and explant culture.

In SCTs, the mRNA expression of HO-1 and HAMP was significantly increased in experimental groups treated with NO (Figure 3). No significant change was seen in the other four genes studied. At the protein level, HO-1 abundance was significantly increased, while hepcidin remained unchanged compared to the control. No change was seen in the other proteins studied, and neither nitrite nor nitrate had any effect on mRNA or protein levels for any of the targets.

In placental explants, mRNA levels of HO-1 and TfR were significantly increased in response to NO (Figure 4A,C), and levels of HAMP trended upwards but did not reach statistical significance. The expression of the remaining targets was unchanged (Figure 4). At the protein level, similar to SCTs, HO-1 abundance was increased in response to NO (Figure 4G), but no significant change was seen in the rest of the proteins studied. Likewise, neither nitrite nor nitrate had a significant effect on mRNA or protein levels in explant samples.

The rates of NO production and consumption, as well as the yield of products of NO metabolism, are influenced by O_2_ levels [8] and may change greatly under conditions of tissue hypoxia which are common to many placental pathologies. Thus, experiments to observe the formation of FeNOs in response to exposure to NO or NOx were also carried out in SCTs cultured under hypoxic conditions. NO metabolism in hypoxic cell culture resembled that observed under normoxic cell culture conditions (Figure 2 and Figure 5). FeNO formation was increased in response to NO compared to the untreated controls, and the FeNO formation could not be attributed to the presence of nitrite or nitrate (Figure 5C). Similar to results under normoxia, mRNA expression levels of HO-1 and HAMP were increased under hypoxic conditions in response to NO (Figure 6A,B), while no change in mRNA expression was observed in the rest of the genes studied. At the protein level, exposure to NO under hypoxic conditions had no significant effect on the protein abundance of any of the iron-handling target proteins.

## 3. Discussion

Iron nitrosyl complexes (FeNOs) have the potential to act as either sinks or stores of NO, thus modulating the amount of both free NO and iron within the cell. While recent discoveries indicate FeNOs play a physiological role in the placenta [17,18,20], the extent to which FeNO formation might influence intracellular iron homeostasis in the placenta is unknown. In this study, we showed that exposure of syncytiotrophoblast cells and villous placental explants to NO at concentrations below the threshold of toxicity results in FeNO formation. We also demonstrated that NO exposure can alter gene expression and protein levels of HO-1, hepcidin, and TfR, thus potentially altering iron handling.

Iron can be present within the cell in many forms, including heme-bound, iron–sulfur clusters, iron oxide within ferritin, or free ferrous or ferric iron. Free NO radicals can bind and react with most of these forms of iron, resulting in the formation of stable FeNO complexes that can preserve the bioactivity of NO [10,21]. Using chemiluminescence methods that are selective for FeNOs [22], we observed that the exposure of both cultured syncytiotrophoblasts and villous to ~250 nM NO for 24 h resulted in detectable levels of intracellular FeNOs. Notably, the FeNOs were not observed in parallel cultures exposed to nitrite or nitrate, indicating that the FeNO formation was derived from NO, not from the presence of its oxidation products nitrite or nitrate. It is worth noting that, in syncytiotrophoblasts, the measurement of total NOx (Figure 2B) was 10-fold greater than that of FeNOs (Figure 2C) for the NO-treated but not nitrite- or nitrate-treated cells. As such, the NOx signal depicted in Figure 2B is not likely to be nitrite or nitrate and suggests the presence of an NO metabolite that is retained within the cells despite the three washes with neat buffer that took place prior to homogenization. The most likely candidates for this signal would be nitrosothiols or FeNOs that are somehow not detectable by the PBS/FeCN method used for selective measurement of FeNOs (Figure 2C), possibly due to the degradation of the FeNO species during sample homogenization.

Approximately 90% of iron in the body is contained within heme, and heme oxygenase plays an essential role in iron homeostasis by recycling heme iron. HO-1 expression is highly inducible in response to a number of stimuli, including oxidative stress, inflammation, and hypoxia [23]. In the human and mouse placenta, HO-1 is expressed exclusively in trophoblasts [24,25], where it contributes to iron handling. NO has been reported to upregulate HO-1 gene expression [26,27] and also to stabilize HO-1 mRNA [26,28]. The results of the current study are consistent with these previous reports, as we observed increases in both mRNA and HO-1 protein levels in both SCTs and explants in response to NO exposure. The implications of this finding are that elevated NO levels may serve to upregulate heme iron recycling in the cytotrophoblasts, potentially making more iron available for the fetus, although correlations between placental HO-1 activity and fetal iron availability have not been established [25]. In addition to iron handling, HO-1 has also been implicated in the process of placental implantation in the first trimester, although reports are conflicting, with one report finding decreased cytotrophoblast invasion following the inhibition of HO activity [29] and another finding that the upregulation of HO-1 activity suppresses invasion [30]. Thus, it is possible that elevated levels of NO during the first trimester may also alter placental implantation via an HO-1-dependent pathway, which would be of relevance to preeclampsia since HO-1 levels are elevated in preeclamptic placental tissue [25,31,32].

Hepcidin, a small peptide hormone coded by the HAMP gene, is known as the master regulator of iron homeostasis. It is mainly produced by the liver and serves to regulate plasma iron levels, primarily by suppressing ferroportin levels in the iron-supplying cells of the body such as hepatocytes, macrophages, and enterocytes [33]. Because ferroportin is the primary route of iron export to the fetus, increased actions of hepcidin in the placenta would potentially lead to fetal iron deprivation. Although hepcidin is found in the placenta and produced by syncytiotrophoblasts, there is conflicting evidence regarding its overall role in downregulating iron transport from the mother to the fetus. On one hand, no correlation was observed between concentrations of hepcidin and placental iron transport [34], and a loss of hepcidin expression in the fetal tissues of the placenta did not alter ferroportin levels or iron levels in the placental or fetal serum [4,35], suggesting that placental hepcidin levels are not essential to regulation of placental iron transport. On the other hand, net iron transfer to the fetus is inversely correlated to maternal hepcidin levels, consistent with the idea that maternal hepcidin may suppress ferroportin in the placenta [36]. Furthermore, the addition of hepcidin to cultured trophoblasts results in a decrease in ferroportin expression [37]. In addition, there is a correlation between increased levels of placental hepcidin and fetal growth restriction and small-for-gestational-age babies [37,38], consistent with a role for hepcidin in regulating iron transport to the fetus. Notably, relatively little is known with regard to whether the hepcidin that may act in the placenta is actually produced by the placenta as opposed to being taken up from the maternal or fetal circulation. The current experiments demonstrate that HAMP is indeed transcribed and translated to hepcidin in cultured syncytiotrophoblasts. The presence of NO resulted in an increase in the levels of HAMP mRNA but did not alter overall hepcidin protein levels. This finding suggests that hepcidin protein abundance may be regulated at the level of translation (e.g., interfering RNA such as miRNA or siRNA) or post translationally (e.g., increased rates of hepcidin degradation and elimination) in a way that abrogates the effects of NO on HAMP transcription. The mechanism of NO-mediated upregulation of HAMP transcription was not investigated in the current studies but may be via the upregulation of the bone morphogenic protein and SMAD signaling pathway, which is known to be the primary determinant of hepcidin expression [39] and is also upregulated in response to the presence of NO from donors [40]. It is worth noting that the upregulation of HAMP transcription in response to NO treatment was noted in SCTs but not in placental explants. This may be due to the fact that explants are comprised of a variety of different cell types, many of which may be less responsive than syncytiotrophoblasts to NO, thereby diluting the syncytiotrophoblast-specific upregulation of HAMP to undetectable levels in whole-explant homogenates. Further experiments with single-cell RNA assessments are needed to assess this possibility.

Fetal iron supply relies on uptake of transferrin-bound iron from the maternal circulation across the apical membrane of the trophoblast cells [41]. This clathrin-mediated endocytotic process is initiated by binding of transferrin to TfR present on the plasma membrane. Iron is then released from the endocytotic vesicle into the cytosol by DMT1, although evidence suggests that other transporters also participate in this process [42,43]. The export of iron from the trophoblast across the basolateral membrane to the fetal circulation is facilitated by ferroportin [44]. In most tissues studied, including trophoblasts [4], levels of TfR, DMT1, and ferroportin can be regulated by iron regulatory protein (IRP). In the event of IRP activation by iron deficiency, IRP binds to the 5′ UTR of TfR and DMT mRNA, thereby stabilizing it and increasing protein translation. Conversely, IRP can bind to the 3′ UTR of many transcripts, including ferroportin, thereby decreasing its translation to protein [4]. Thus, IRP activity represents a target with the potential for affecting the levels of several components of iron homeostasis. NO has been found to modulate IRP levels and activity [11,12,13,15,16] by either interacting directly with its iron–sulfur cluster to activate the RNA-binding activity of IRP or secondary to the modulation of the intracellular labile iron pool by reactions with NO [15]. These previously described effects of NO on IRP appeared to be of minimal consequence in the current experiments with cultured SCTs and explants since although TfR mRNA levels were increased in explants exposed to NO, other mRNA levels and protein levels of TfR, DMT1, and ferroportin were not affected. To our knowledge, the observation of increased TfR mRNA levels in explants exposed to NO has not been previously described in placental tissue. The fact that only mRNA and not protein levels were elevated in the explants suggests there may be other epigenetic or post-translational points of regulating TfR protein concentrations in the placenta. Notably, the increase in TfR mRNA was not observed in syncytiotrophoblasts, raising the possibility that the increase observed in the explants may have been due to the effects of NO on other cell types within the explants or that the interaction of the syncytiotrophoblasts with the other cell types may be required for the effects of NO to be observed. Notably, NO has also been found to regulate levels of DMT1 [45] and ferroportin [33] via IRP-independent pathways in other tissue types, none of which appeared to be altered significantly in the current experiments. Furthermore, as discussed above, it is notable that ferroportin levels were unchanged by NO despite changes in HAMP and hepcidin, which is known to reduce ferroportin levels in many other tissue types.

As a free radical, NO can participate in a variety of different biochemical reactions within the cell. Under normoxic conditions, a major portion of these reactions involve molecular O_2_ or reactive oxygen species derived from O_2_. Thus, the half-life of NO, the types of reactions that consume it, and the range of products that result from its metabolism are all influenced by decreases in O_2_ availability [8]. Given much evidence that hypoxia is prevalent in preeclamptic placentas, the current experiments were repeated with SCT cell cultures carried out at 2% O_2_ (PO_2_ approximately 13 mmHg). Consistent with previous work by others (see review by [8]), hypoxia did appear to prolong the half-life of NO disappearance, as evidenced by the ~2-fold greater concentrations of free NO in the hypoxic culture media (Figure 1). However, the effects of hypoxia alone on mRNA and protein levels of the targets studied were minimal, and it did not significantly alter the responses to NO that were observed in the normoxic experiments (Figure 6). While there is no evidence in the current results to suggest that the effects of NO on iron handling in trophoblasts are significantly affected by hypoxia, it is a limitation of the study that both O_2_ and NO levels were changed in our hypoxia experiments. For example, cellular metabolic responses to hypoxia may have blunted the effects of NO such that the two-fold greater concentrations did not result in greater effects on the iron-handling targets. Further work that evaluates responses to a broader range of O_2_ concentrations with NO concentrations held constant would be important to explore this possibility further, particularly in the context of diseases where placental hypoxia is thought to play a prevalent role, such as preeclampsia.

Our previous work establishing that the placenta can produce significant amounts of FeNOs [18,20], particularly in preeclamptic placentas [17], together with the understanding that NO can cause intracellular iron deficiency by scavenging iron for the production of FeNOs [5], raised the question of whether or not NO may cause aberrant iron handling in syncytiotrophoblasts. The results of the current studies indicate that NO, at concentrations below the threshold of cellular toxicity, can impact the iron-handling machinery within the placenta so as to increase iron availability via increased TfR and HO-1 levels. Whether the effects of NO on these iron handling targets result in a net increase in placental iron content or transfer of iron from the mother to the fetus remains to be determined. Further work is also needed to assess whether NO affects changes in distribution of iron within the various intracellular pools, such as ferritin, and to characterize what proteins are involved in FeNO formation.

## 4. Materials and Methods

### 4.1. BeWo Cell Culture

Culture of BeWo cells, a human cell line that is commonly-studied for its phenotypic similarities to placental trophoblasts (ATCC, CCL-98™), was carried out as previously described [18]. Briefly, cells were cultured in Ham’s F12 (Kaighn’s) growth medium (Gibco^TM^, Grand Island, NY, USA) enriched with 10% fetal bovine serum (Gibco^TM^, Grand Island, NY, USA) and 1% Penicillin/Streptomycin (Sigma-Aldrich, Saint Louis, MO, USA), according to the manufacturer’s protocol. Cells were routinely maintained in T75 flasks (Southern Labware, Cumming, GA, USA) at pH 7.4 under 5% CO_2_ and 95% humidity at 37 °C and passaged upon reaching 70 to 80% confluency. To investigate the effect of NO on gene expression of iron handling targets in cultured syncytiotrophoblasts, cells were allowed to reach 50 to 60% confluency and induced to differentiate with 50 μM forskolin (FSK) (Cayman Chemical, Ann Arbor, MI, USA) for 48 h. After 48 h, FSK-containing media was removed, and fresh FSK-free media was added along with respective study treatments for another 24 h.

### 4.2. Placental Explant Culture

Placentas were collected and processed as previously described [18]. Briefly, term placentas (≥37 and ≤42 weeks gestation) were collected within 30 to 60 min following vaginal or Caesarean section birth from patients following complication-free pregnancies. Placentas were weighed without the umbilical cord and after separation from the amniotic chorionic membrane. Villous tissue was randomly sampled from multiple placental cotyledons by avoiding the chorionic plate and trimming off the decidual layer. The remaining villous tissue from different placental sections was washed in 1x Dulbecco’s phosphate-buffered saline (DPBS) (Gibco^TM^, Grand Island, NY, USA) (pH = 7.4) to remove excess blood, then dissected into 5 to 15 mg villous tissue explants randomly selected and placed into 100 mm culture dishes (Southern Labware) with at least 0.5mL of culture media per explant. The culture media was complete Iscove’s modified Dulbecco’s medium (IMDM) with 10% FBS (Gibco^TM^, Grand Island, NY, USA) and 1% each of L-glutamine and penicillin/streptomycin mixture (Sigma-Aldrich, St. Louis, MO, USA).

### 4.3. NO and Hypoxia Treatments

Cell and explant cultures were exposed to NO by applying the NO donor (Z)-1-[N-(2-aminoethyl)-N-(2-ammonioethyl)amino]diazen-1-ium-1,2-diolate (DETA), which degraded to release NO with a half-life of approximately 19 h [46]. This concentration was selected based on dose-response experiments that demonstrated this level of DETA had no effect on culture viability, as described in the Results. To control for the oxidation of NO derived from the DETA, which would result in an accumulation of nitrite and nitrate in the culture media, control groups received 250 μM NaNO_2_ (Sigma-Aldrich, St. Louis, MO, USA) or 250 μM NaNO_3_ (Sigma-Aldrich, St. Louis, MO, USA). At the end of the 24 h study period, samples of media and cells or explants were collected and immediately processed or stored at −80 °C until further processing.

To test for the effects of hypoxia, immediately after addition of NO, nitrite, or nitrate BeWo cell cultures were placed in a hypoxic incubator at at 37 °C and and 95% humidity in 2% O_2_ and 5% CO_2_ with a balance of N_2_ for the duration of the 24 h NO treatment period. These experiments were not carried out in placental explants since the decrease in PO_2_ under these experimental conditions leads to slowed diffusion of O_2_ into the interior regions of the explant such that the results are confounded by cells that must respond to relative anoxia.

### 4.4. Cell Viability

To determine cell viability in response to our treatment, cells were treated with alamarBlue™ Cell Viability Reagent (Thermo Fisher Scientific, Waltham, MA, USA) solution following manufacturer’s instructions. Briefly cells were seeded at 20,000 per well in 96-well culture plates in triplicate and allowed to attach and stabilize overnight. The medium was then changed to 200 μL of 50 μM FSK containing fresh medium per well to induce differentiation of BeWo cells for 48 h. After 48 h, medium was changed to FSK-free medium containing the assigned NO or NOx treatment at (increasing concentrations of up to 2mM DETA, and 250 μM nitrite or 250 μM nitrate) 200 μL per well for 24 h. At 24 h, treatment medium was removed, wells were washed with 1×DPBS, and fresh medium with 10% Alamar Blue solution was added to the wells for 6 h. The absorbance at 570 nm and 630 nm (as background optical density) was then measured with a BioTek Synergy HT Multi-Mode Microplate Reader S1AFR (Biotek, Santa Clara, CA, USA).

### 4.5. Cell and Placental Explant Processing for NO and NOx Measurements

Following the 24 h incubation period, culture media was collected for NOx assays. Cells were then washed with fresh 1x DPBS (Gibco^TM^, Grand Island, NY, USA) three times to remove any residual DETA, nitrite, or nitrate prior to being scraped off the culture flasks and diluted in 500 µL of fresh 1×DPBS. Cells were lysed with three freeze-thaw cycles and then injected into the purge vessel to quantify NOx concentrations as previously described [18].

Explants were homogenized on ice using a TissueRuptor homogenizer (Qiagen, Redwood City, CA, USA), and the homogenate was centrifuged at 1000× *g* for 15 min at 4 °C to remove tissue debris. The supernatant was then immediately assayed for NOx and FeNOs by ozone-based chemiluminescence with either triiodide or ferricyanide-based assays as described below.

### 4.6. NO and NOx Measurement

Concentrations of NO and its metabolites (NOx) in media, cells, and placental explants from each treatment group were determined by ozone-based chemiluminescence (NOA 280i; Sievers, Boulder, CO, USA) using various purge vessel reagents that facilitate discrimination between FeNOs (dinitrosyl iron complexes and heme nitrosyls) and other NO metabolites such as nitrite, nitrate, and nitrosothiols as previously described [22]. In brief, the concentration of free NO derived from DETA in the culture media was assayed by injecting 100 uL of sample into a purge vessel containing 1x DPBS (Gibco^TM^, Grand Island, NY, USA) continuously sparged with argon which entrained the NO into the analyzer. Triiodide reagent (I_3_^−^) was used (assay referred to as I_3_^−^) to determine the concentrations of all NOx species, consisting mainly of nitrite, nitrosothiols, N-nitrosyls, and FeNOs but not nitrate. Nitrate concentration was determined by incubating the sample for 45 min at 37 °C with bacterial nitrate reductase (Sigma-Aldrich, St. Louis, MO, USA), flavin adenine dinucleotide (Sigma-Aldrich, St. Louis, MO, USA), and NADPH (Sigma-Aldrich, St. Louis, MO, USA), followed by injection in triiodide reagent (referred to as NiR/I_3_). For selective measurement of FeNOs without nitrite and nitrosothiol interference, the purge vessel contained 0.2 M potassium ferricyanide in PBS buffer at pH 7.4. All assays with PBS included 100 µL of 1-octanol as an anti-foam agent when biological samples were being measured.

### 4.7. qPCR

Total RNA was extracted from placental explants or cultured cells using the Invitrogen™ PureLink™ RNA Mini Kit (Life Technologies, Carlsbad, CA, USA). The Nanodrop 2000 Spectrophotometer (Thermo Fisher Scientific, Waltham, MA, USA) was used to measure quality and quantity of RNA for cDNA processing. cDNA synthesis was carried out with a cDNA Synthesis Kit (Bio-Rad Laboratories, Foster City, CA, USA) per manufacturer’s instructions using the thermocycler Mastercycler (Eppendorf, Enfield, CT, USA). Product cDNA was reacted with SsoAdvanced™ Universal SYBR^®^ Green Supermix, according to the manufacturer’s protocol in a CFX Connect Real-Time PCR Detection System (Bio-Rad Laboratories, Foster City, CA, USA). The relative expression of the target mRNA was determined using YWHAZ as a reference gene. The primers used were: YWHAZ (NM_001135701.2); ACT TTT GGT ACA TTG TGG CTT CAA (forward); and CCG CCA GGA CAA ACC AGT AT (reverse). Transferrin Receptor (NM_003234.3); TGC TGT GAT CGT CTT TTT CTT GA (forward); and TCA TCC CAA TAT AAG CGA CG TG (reverse). DMT1 (NM_001174127.2); CAC CGT CAG TAT CCC AAG GT (forward); and CAT GTC TGA GCC GAT GAT AGC (reverse). FPN (NM_014585.6); TTA CCA GAA AAC CCC AGC TC (forward); and CAG GGG TTT TGG CTC AGT AT (reverse). HAMP (NM_021175.4); TCC CCA TCT GCA TTT TCT GCT (forward); and GGC AGG TAG GTT CTA CGT CT (reverse). HO-1 (NM_000625.4); AGA CGG CTT CAA GCT GGT GA (forward); and TCT CCT TGT TGC GCT CAA TC TC (reverse). Standard curves (initial amount of cDNA versus Ct values) were tested for each set of primers, demonstrating that for the similar range of total cDNA amplification, the efficiency of target genes and reference gene (YWHAZ) were comparable. All samples were run in triplicate. qPCR product from each of the primers was run on an agarose gel to verify amplification of the desired length product. Relative expression analysis was conducted by the ΔΔCt method and data is presented as fold change normalized to YWHAZ.

### 4.8. Protein Isolation and Quantification

Protein isolation from cells was conducted using Cell Lytic (Sigma-Aldrich, St. Louis, MO, USA) with cOmplete™, Mini Protease Inhibitor Cocktail (Sigma-Aldrich, St. Louis, MO, USA) according to the manufacturer’s instructions. Protein isolation from explants was carried out using T-PER™ Tissue Protein Extraction Reagent (Thermo Fisher Scientific, Waltham, MA, USA) according to the manufacturer’s instructions. Isolated protein from cells and placental explants was quantified using BCA Assay concentrate (Bio-Rad Laboratories, Foster City, CA, USA) with BSA as standard.

### 4.9. Western Blot

For quantification of specific protein expression of iron handling targets the automated, gel-free Western blot system, Simple Western™ was used (Protein Simple, San Jose, CA, USA). The antibodies used were actin (catalog #3700S) and HO-1 (catalog #70081S) from Cell Signaling Technologies (Danvers, MA, USA); transferrin (catalog #NB500-418), ferroportin (catalog #NBP1-21502), and transferrin receptor (catalog #NBP2-32945) from Novus Biologicals (Centennial, CO, USA); DMT (catalog #PA5-35136) from Thermo Fisher Scientific (Waltham, MA, USA); and hepcidin (catalog #SAB1405228) from Sigma-Aldrich (St. Louis, MO, USA).

### 4.10. Statistical Analysis

Statistical analysis was performed using GraphPad Prism 9 (GraphPad Software, San Diego, CA, USA). Data are presented as mean ± standard deviation (SD). Statistical comparison between multiple groups was carried out using one-way ANOVA with Tukey’s multiple comparison test and significance was determined by *p* < 0.05.

## Figures and Tables

**Figure 1 ijms-24-05887-f001:**
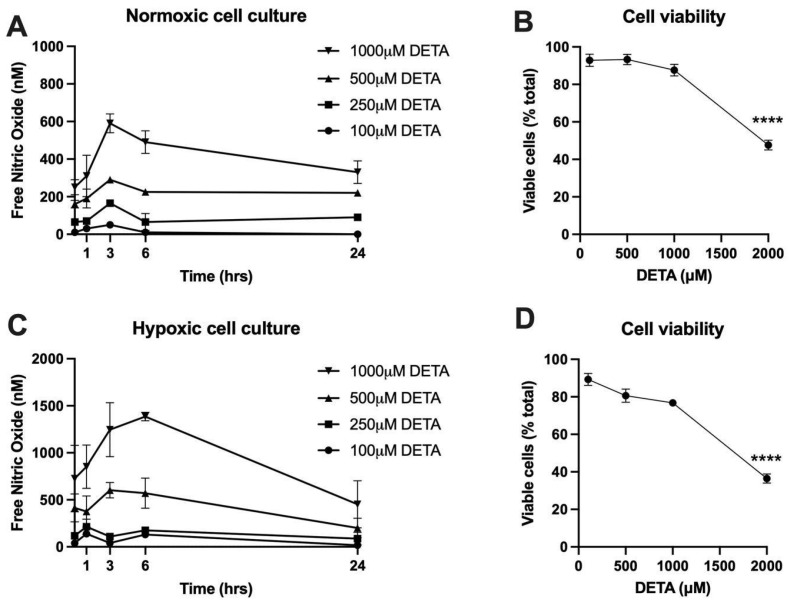
Nitric Oxide (NO) concentrations and cell viability over 24 h study period. (**A**) NO production from different concentrations of DETA in normoxic cell culture of syncytialized BeWo cells. (**B**) BeWo cell viability in response to increasing DETA concentrations under normoxia. (**C**) NO production from different concentrations of DETA in hypoxic (2% O_2_) cell culture of syncytialized BeWo cells. (**D**) BeWo cell viability in response to increasing DETA concentrations under hypoxia, mean ± SD, **** *p* < 0.0001 compared to no DETA.

**Figure 2 ijms-24-05887-f002:**
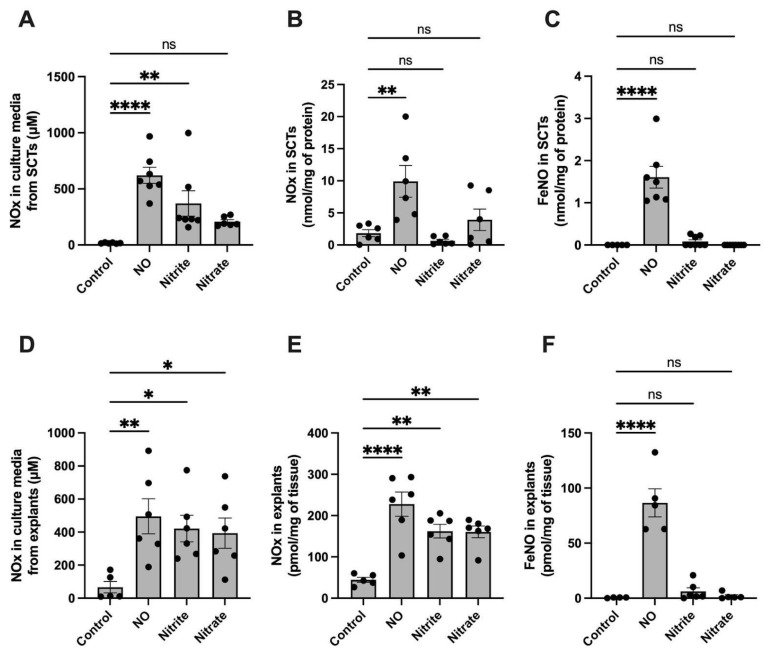
NO metabolites and FeNO formation in cultured BeWo cells and villous explants after 24 h of treatment with NO (as 500 μM DETA), nitrite (250 μM), or nitrate (250 μM). (**A**) Total NOx (NO, nitrite, nitrite, FeNOs, and SNOs) in normoxic cell culture media of syncytialized BeWo cells (SCTs) following 24 h of treatment with NO, nitrite or nitrate. (**B**) NOx (NO, nitrite, FeNOs, and SNOs but not nitrate) found in homogenized syncytialized BeWo cells following 24 h treatment with NO, nitrite, or nitrate. (**C**) FeNO levels found in syncytialized BeWo cells following 24 h treatment with NO, nitrite, or nitrate. (**D**) Total NOx in culture media of human placental explants following 24 h of treatment with NO, nitrite, or nitrate. (**E**) TotalNOx found in human placental explants following 24 h treatment with NO, nitrite, or nitrate. (**F**) FeNO levels found in placental explants following 24 h treatment with NO, nitrite, or nitrate. FeNO formation is significantly increased by the presence of NO but not nitrite or nitrate. Data are reported from at least three independent experiments as mean ± SD, * *p* < 0.05, ** *p* < 0.01, **** *p* < 0.0001, ns = not significant.

**Figure 3 ijms-24-05887-f003:**
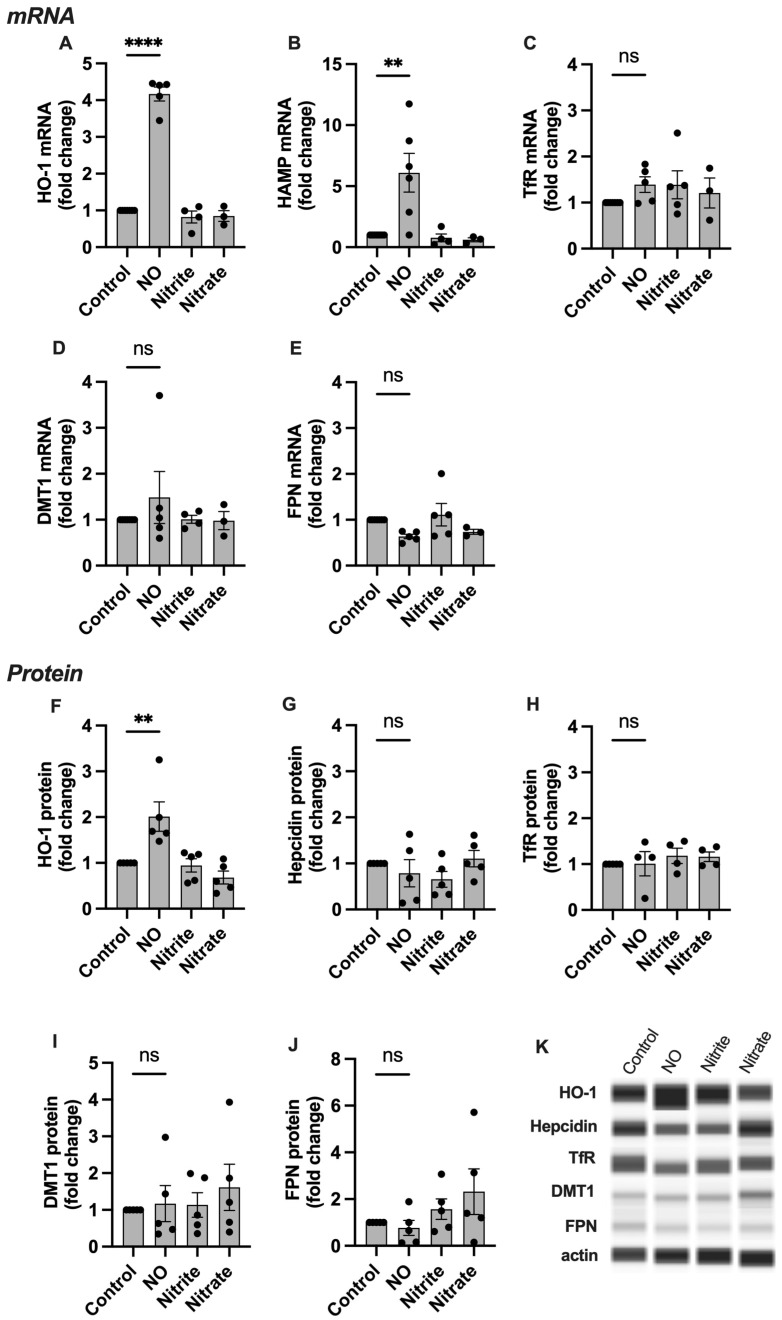
The effect of NO on iron handling genes in syncytialized BeWo cells. (**A**–**E**), qPCR results, and (**E**–**J**) protein analyses. Representative blots are shown in (**K**). HO-1 and HAMP mRNA levels are significantly increased in the group treated with NO compared to the untreated controls. Protein abundance of HO-1 is significantly increased in the group treated with NO compared to the untreated controls. No significant change in expression is seen in the rest of the genes studied. HO-1: heme oxygenase 1, HAMP: hepcidin antimicrobial peptide, TfR: transferrin receptor, DMT1: divalent metal transporter 1, FPN: ferroportin 1. Data are reported from at least three independent experiments as mean ± SD, ** *p* < 0.01, **** *p* < 0.0001, ns = not significant.

**Figure 4 ijms-24-05887-f004:**
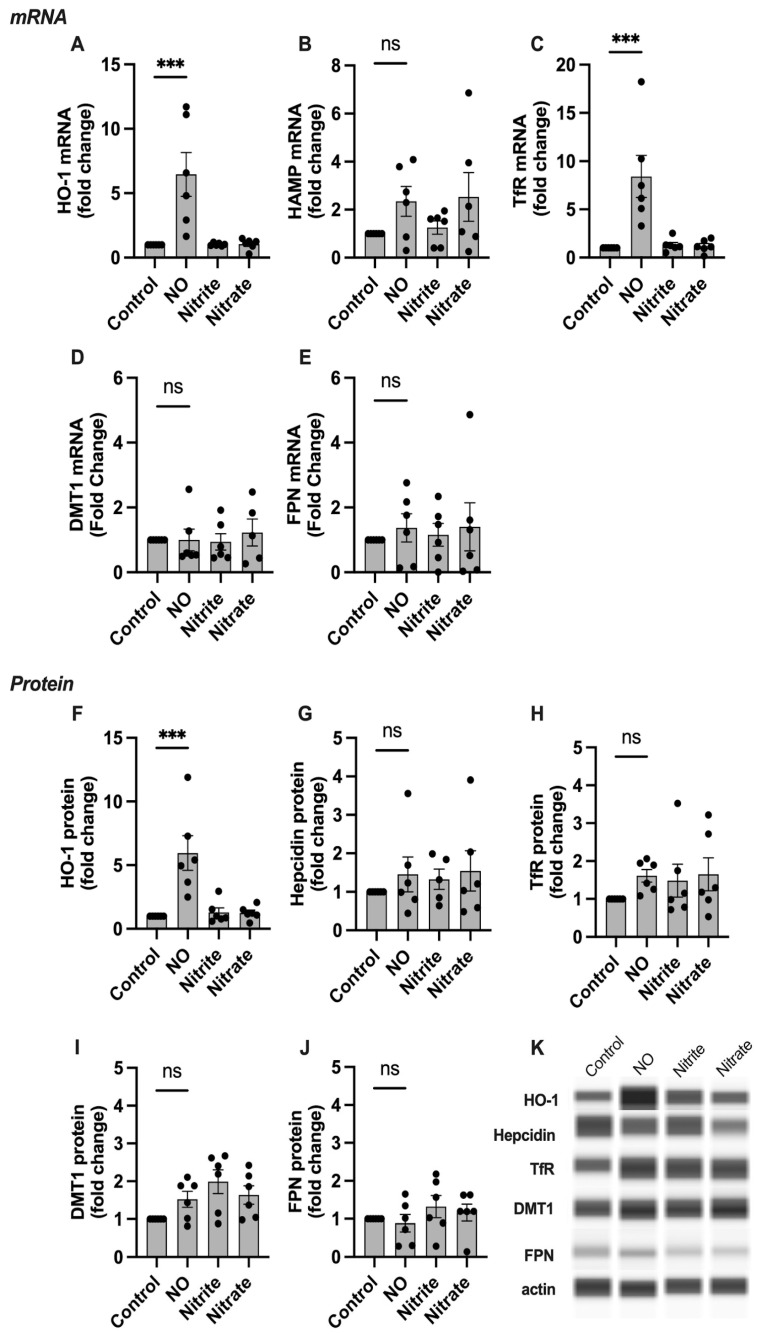
The effect of NO on iron handling gene expression in human placental villous explants. (**A**–**E**) qPCR results, (**F**–**J**) protein analyses. Representative blots are shown in (**K**). HO-1 and TfR mRNA levels are significantly increased in the group treated with NO compared to the untreated controls. Protein abundance of HO-1 is significantly increased in the group treated with NO compared to the untreated controls. No significant change in expression is seen in the rest of the genes studied. HO-1: heme oxygenase 1, HAMP: hepcidin antimicrobial peptide, TfR: transferrin receptor, DMT1: divalent metal transporter 1, FPN: ferroportin 1. Data are reported from at least three independent experiments as mean ± SD, *** *p* < 0.001, ns = not significant.

**Figure 5 ijms-24-05887-f005:**
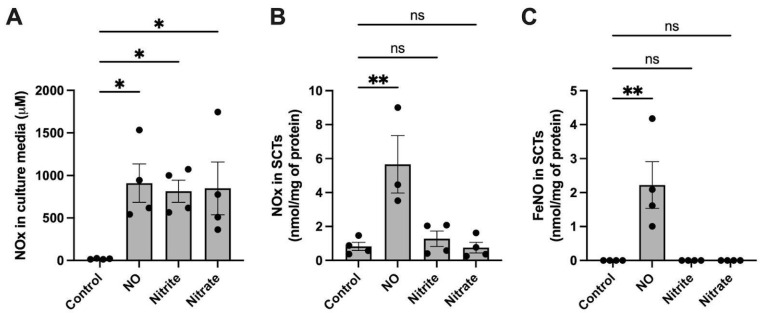
NO metabolism and FeNO formation in BeWo cell culture after 24 h of treatment with NO (as 500 μM DETA), nitrite (250 μM), or nitrate (250 μM) under hypoxic conditions. (**A**) Total NOx in hypoxia cell culture media of syncytialized BeWo cells (SCTs) following 24 h of treatment. (**B**) NOx found in SCTs following 24 h treatment with NO, nitrite, or nitrate under hypoxic conditions. (**C**) FeNO levels found in SCTs following 24 h treatment with NO, nitrite, or nitrate under hypoxic conditions. FeNO formation is significantly increased following exposure to NO but not nitrite or nitrate. Data are reported from at least three independent experiments as mean ± SD, * *p* < 0.05, ** *p* < 0.01, ns = not significant.

**Figure 6 ijms-24-05887-f006:**
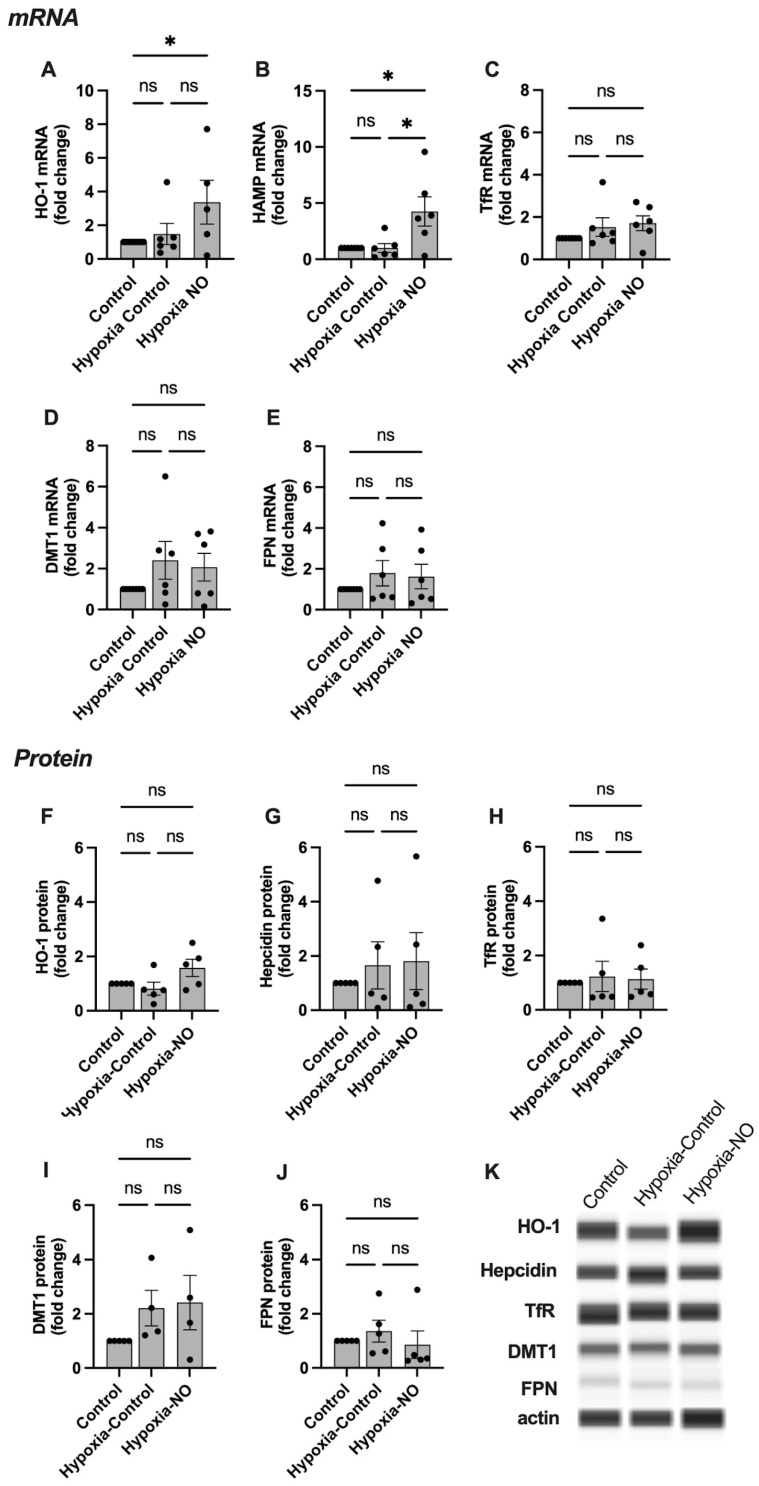
The effect of hypoxia on iron handling genes in response to NO in syncitialized BeWo cells (SCTs). (**A**–**F**) qPCR results and (**G**–**K**) protein analyses. Representative blots are shown in (**K**). Hypoxia alone did not significantly alter mRNA or protein levels of any of the studied iron homeostasis markers compared to normoxic controls. (**A**,**B**) The combination of hypoxia and NO resulted in a significant increase in HO-1 and HAMP mRNA levels, while NO increased HAMP mRNA levels significantly under hypoxic conditions. No significant effect of hypoxia or NO was seen in any of the other targets studied. HO-1: Heme Oxygenase 1, HAMP: Hepcidin antimicrobial peptide, TfR: Transferrin Receptor, TF: Transferrin, DMT1, Divalent metal transporter 1, FPN: Ferroportin 1. Data are reported from at least three independent experiments as mean ± SD, * *p* < 0.05, ns = not significant.

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
