# Peer review of "Nitric Oxide Affects Heme Oxygenase-1, Hepcidin, and Transferrin Receptor Expression in the Placenta"

_ijms, 2023, doi:10.3390/ijms24065887_

Round 1

Reviewer 1 Report

Major:

There is no need for subheading in the Introduction. Remove subheadings and join separate introduction sections.

Line 106: How did authors decide on the specific concentration of NO2? Please clearly explain. Also, be consistent with abbreviation use. Once abbreviated use the abbreviation instead of the complete name.

Line 118: State the exact metabolites.

In the Materials and Methods section the authors describe the NO metabolites assays as having the ability to measure each metabolite separately. Why are authors showing total NOx levels instead of each metabolite in Figure 2?

Lines 139-144. Did authors check for HO-1 activity in both BeWo cells and explants?

Figure 3: western blots: Why are bands of the same protein appearing at different molecular weights?

Figure 3: Authors show an increase of HAMP mRNA levels in response to NO but no change in protein levels. Do authors have an explanation?

Also the increase in HAMP mRNA levels in response to NO in not observed in explants. How do authors explain this? The same question stands for the increase in TFR-1 observed in explants but not in cells.

All this should be discussed in the Discussion section.

Figure 6 western blots. Again bands of the same protein are appearing at different molecular weights.

Discussion section: There is no need to have separate sections. Combine all sections in one.

Discussion lines 297-235. Authors are showing results for TFR-1. However, in the discussion section they refer to TF-R. Authors need to be specific and consistent with name/abbreviation use.

Discussion lines 332-339: This is speculation. Although authors suggest that no effect is expected under hypoxia experiments of explants under hypoxia should also be performed.

Minor:

English language use should be checked throughout the manuscript. For example:

line 24: change was to were.

Abbreviation use should be checked throughout the manuscript. Lines: 223, 267, 297

Reviewer 2 Report

The paper “Nitric oxide affects heme oxygenase-1, hepcidin and transferrin 2 receptor expression in the placenta” by Principe et al. demonstrated that the relationship between some proteins and NO is involved in the pathogenesis of preeclampsia. This manuscript contains very useful data for many researchers. However, the data in this manuscript is too simple and it is difficult to understand the function. All concerns listed below, and the discussion should be rewritten to be more consistent throughout for improving the overall quality.

Introduction: Although this manuscript describes in detail the intracellular behavior of iron, about half of the description is not necessary for this manuscript and should be deleted to make it brief.

In Fig. 4 and Fig. 6, the mRNA increases, but there is no change in the protein level. As for hepcidin, "The presence of NO resulted in an increase in the levels of HAMP mRNA, but did not alter overall hepcidin protein levels, indicating that hepcidin abundance may be controlled at a post transcriptional level in a way that abrogates the effects of NO on HAMP transcription.” However, there are no results to prove this.

Since HO-1 and FeNO seem to be important in this study, it would be easier to understand if the other proteins were basically presented in a simplified form and data on functional changes of the proteins were added.

Reviewer 3 Report

The present manuscript discussed the influence of sub-cytotoxic concentrations of nitric oxide on the key iron regulatory genes and proteins expression in placental syncytiotrophoblasts and villous tissue explants. Overall, this is original study, novel findings have been obtained, the manuscript is generally well written and data are well described. However, there are few concerns that need to be clarified.

Introduction is generally well written and is a good starting point to understand the whole study; however, the role of hypoxia in preeclamptic placentas should be more in-depth explained in Introduction. Besides, at the end of the Introduction section, objectives of the study should be defined instead of summarizing the main results.

Paragraph 2.1. – was cell viability significantly decreased after 500 µM DETA treatment under hypoxic conditions? It seems so in the Figure. Besides, the higher concentrations of NO under hypoxic conditions should be further discussed. The authors stated that ‘hypoxia did appear to prolong the half-life of NO disappearance’ (line 333), but it should refer to some other studies.

In line 116, BeWo cells were first time mentioned and thus should be defined there. Besides, in Figure 2 and in the legend of Figure 6, the abbreviation SCT is mentioned – it should be defined in the text as well, when syncytiotrophoblasts were first mentioned.

There are some other technical issues in the manuscript as well.

It should be oxygen instead of O2, at least when the first time mentioned (although I suggest it to change it everywhere).

Line 59 – L-arginine instead of L-Arginine.

Line 65 – iron has already been mentioned several times, so ‘Fe’ should be earlier in the text.

Line 97 – 37 °C instead of 37 C.

Line 102 – why ‘approximately’ (~ symbol)?

I am not completely sure if the reference style (numbers in the superscript) are in accordance with the journal instructions, so please check that too.

Round 2

Reviewer 1 Report

Authors have addressed all reviewer's comments adequetely.

Author Response

We thank the reviewer for their helpful comments and for taking the time to help us strengthen the manuscript for publication.

Reviewer 2 Report

No response to our last comment. You should reply to the comment.

Author Response

Since HO-1 and FeNO seem to be important in this study, it would be easier to understand if the other proteins were basically presented in a simplified form and data on functional changes of the proteins were added.

We apologize for inadvertently neglecting to include our response to this point in our initial resubmission! We are in agreement with the reviewer that the HO-1 and FeNO findings are of particular interest. However, the goal of the current study was to assess the overall effects of NO and FeNO formation on the cellular iron regulatory proteins. Now that we have discovered there is indeed an interaction between NO and some of the regulators of Fe homeostasis, we are awaiting funding for future work that will focus on activity levels, overall intracellular Fe levels, and the balance between heme and non-heme Fe pools and CO and biliverdin production, in which HO-1 plays a key role.
